# A Social Network Analysis of Twitter Data Related to Blood Clots and Vaccines

**DOI:** 10.3390/ijerph19084584

**Published:** 2022-04-11

**Authors:** Wasim Ahmed, Josep Vidal-Alaball, Josep M. Vilaseca

**Affiliations:** 1Management School, University of Stirling, Stirling FK9 4LA, UK; wasim.ahmed@stir.ac.uk; 2Health Promotion in Rural Areas Research Group, Gerència Territorial de la Catalunya Central, Institut Català de la Salut, 08772 Sant Fruitós de Bages, Spain; 3Unitat de Suport a la Recerca de la Catalunya Central, Fundació Institut Universitari per a la Recerca a l’Atenció Primària de Salut Jordi Gol i Gurina, 08772 Sant Fruitós de Bages, Spain; 4Faculty of Medicine, University of Vic-Central University of Catalonia, 08500 Vic, Spain; josepmaria.vilaseca@umedicina.cat; 5Primary Care Service, Althaia Xarxa Assistencial Universitària de Manresa, 08243 Manresa, Spain

**Keywords:** COVID-19, Twitter, blood clots, social media, clots

## Abstract

After the first weeks of vaccination against SARS-CoV-2, several cases of acute thrombosis were reported. These news reports began to be shared frequently across social media platforms. The aim of this study was to conduct an analysis of Twitter data related to the overall discussion. The data were retrieved from 14 March to 14 April 2021 using the keyword ‘blood clots’. A dataset with *n* = 266,677 tweets was retrieved, and a systematic random sample of 5% of tweets (*n* = 13,334) was entered into NodeXL for further analysis. Social network analysis was used to analyse the data by drawing upon the Clauset–Newman–Moore algorithm. Influential users were identified by drawing upon the betweenness centrality measure. Text analysis was applied to identify the key hashtags and websites used at this time. More than half of the network comprised retweets, and the largest groups within the network were broadcast clusters in which a number of key users were retweeted. The most popular narratives involved highlighting the low risk of obtaining a blood clot from a vaccine and highlighting that a number of commonly consumed medicine have higher blood clot risks. A wide variety of users drove the discussion on Twitter, including writers, physicians, the general public, academics, celebrities, and journalists. Twitter was used to highlight the low potential of developing a blood clot from vaccines, and users on Twitter encouraged vaccinations among the public.

## 1. Introduction

Since the beginning of the COVID-19 pandemic, caused by the SARS-CoV-2 virus 2019 [1], it was reported by media outlets that one of the features of COVID-19 is thrombosis (popularly known as blood clots) [2]. Several studies confirmed the observation made by initial publications on thrombotic complications in COVID-19 patients that a diagnosis of thromboembolism per se was associated with a more complicated in-hospital clinical course, a higher incidence of admittance to the intensive care unit and higher all-cause mortality [3]. However, the early identification of patients at risk of thrombosis is still a challenge. The protocols used to treat COVID-19 may differ across Europe. The role of heparin in the prevention of thrombosis and acute embolism seems to be of importance in severely ill COVID-19 patients [4]. In Spain, hospitalised patients are usually treated with heparin, whereas homecare patients are not.

After the first weeks of vaccination against SARS-CoV-2, several cases of acute thrombosis were reported. These cases were mainly associated with the AZD1222 (Oxford-AstraZeneca) and Janssen COVID-19 (Johnson & Johnson) vaccines. The events had a great impact on the media, and a debate emerged on the safety of COVID-19 vaccines in general. On 10 March 2021, Austria suspended the use of a batch of AstraZeneca vaccines after one person had multiple thromboses diagnosed and died ten days after vaccination [5]. The next day, eight European countries suspended the use of AstraZeneca’s vaccine [6]. Italy, France, Germany and Spain suspended the use of the vaccine on 15 March. This cascade of European countries ceasing the vaccination programme prompted the pharmacovigilance agencies to react. Hence, on 18 March, both the British Medicine and Healthcare Regulatory Authority (MHRA) and the European Medicines Agency (EMA) stated independently that the AstraZeneca vaccine was safe and effective and that there was no evidence of an association between the vaccine and the reported cases of blood clots [7].

The national authorities did not feel fully satisfied with these statements. On 25 March, the EMA published a note saying that “the committee confirmed that the vaccine is not associated with an increase in the overall risk of blood clots and that the benefits of the vaccine in combating the still widespread threat of COVID-19 continue to outweigh the risk of side effects”. The committee recommended including more information and advice for healthcare professionals and the public in the vaccine’s product information [8]. Immediately, AstraZeneca modified the product’s information and published a direct communication to healthcare professionals warning of the notified cases of thrombosis after vaccination and reminding them of the signs and symptoms of thrombosis [9]. However, the European Commission decided to not purchase the vaccine from 14 April 2021 [10]. 

Governments and public health authorities have noted the potential of large-scale vaccination programmes as a way out of the COVID-19 pandemic. Reports of blood clots may potentially deter certain individuals from obtaining a vaccine and can endanger the aim of controlling the pandemic. Although vaccination has been shown throughout history to have great benefits, vaccine hesitancy can be a major public health problem, and the World Health Organisation (WHO) has considered it one of the top ten threats to global health [11,12]. 

It is well known that anti-vaccine movements have actively used social media in order to spread rumours about the alleged dangers of vaccines [13]. Since the beginning of the COVID-19 pandemic, discussions around vaccines have been taking place on social media [14], and lately discussions around blood clots have taken place frequently on Twitter. From a public health standpoint, it is important to gain an understanding of the types of discussion taking place on social media in order to be able to identify and counter the spread of negative messages quickly and effectively. The aim of this study is to analyse Twitter discussions around thrombosis-related events associated with COVID-19 vaccination. Moreover, the study aims to understand the key narratives, groups, and content within this network, the negative or positive perceptions surrounding vaccination and how the news about possible blood clots affects these perceptions. Our results are likely to be of interest to health organisations and governments around the world. 

### Previous Research on COVID-19 and Social Media

A rapid literature search on COVID-19 and social media retrieves many articles. When limiting this search to the use of Twitter, the number of papers reduces, and when focusing on thrombosis, only six papers are found, of which only a couple appear relevant. One manuscript highlights that the use of Twitter enables interested health professionals and the public to stay informed; however, the authors do not use Twitter as a means of retrieving information on the perceptions of patients and the general population [15]. Another study also uses Twitter as a search engine to identify scientific papers related to thrombosis, but not for social network analysis [16]. No previous research has been found related to the perceptions of patients regarding COVID-19 vaccines and blood clots.

## 2. Materials and Methods

This study retrieved data from the Twitter-search application programming interface (API) using Twitter Archiving Google Sheets (TAGS). This API is likely to provide enough tweets if the time duration is short and it involves the use of few keywords, which was the case in our study. Previous research has found that for a single-keyword search, the search API may be able to retrieve up to 79% of all tweets when compared to the Firehose (which provides all tweets) [17]. Data were retrieved from 14 March to 14 April using the keyword ‘blood clots’. This keyword was selected in place of medical terms, such as thrombosis, due to the wider use of the term ‘blood clots’ during this time. The English language was the focus of this study because it is the most commonly used language in which tweets are sent. 

A dataset with *n* = 266,677 tweets was retrieved, and a systematic random sample of 5% of tweets (*n* = 13,334) was entered into NodeXL for further analysis. We used a similar methodology to previous research [18,19]. Social network analysis, which is useful in highlighting the key groups of discussion within a network, was used to analyse the data. This study uses the network shapes and structures defined in a previous empirical study [20]; readers new to social network analysis may wish to consult this overview and classification of social media networks.

A network graph was produced by analysing relationships between Twitter accounts such as tweets, retweets and mentions; accounts that conversed with each other were assigned into distinct groups within the network. More specifically, the Clauset–Newman–Moore algorithm was used to identify the key communities within the discussion [21]. Influential users were identified by using the betweenness centrality algorithm, designed to rank users according to their position in a network [22]. The study only made use of public data, which were analysed in aggregate. 

## 3. Results

There were 13,874 Twitter users within our dataset. The dataset consisted of *n* = 9900 retweets, *n* = 1952 tweets, *n* = 1293 replies, *n* = 1261 mentions and *n* = 765 mentions in retweets. This highlights that the network had a high retweet ratio and was a broadcast network. The social network analysis of the data revealed that the discussion around blood clots was formed across several communities. Figure 1 is a diagrammatic representation of Twitter users. Each circle represents Twitter users. The larger circles represent users who were influential (using the betweenness centrality measure), and the smaller dots represent users who were less influential within the network. The groups are presented from left to right by size. The four largest groups are labelled (Group 1, Group 2, Group 3 and Group 4). 

The figure highlights that there were four key groups of users, followed by a number of smaller pockets of discussion. Groups 1 and 2 are termed ‘broadcast’ groups, which had an impact on the entire network. Discussions on social media can often be led by a number of ‘power’ users who are retweeted in large frequencies. The shapes in Groups 1, 2 and 4 reveal a number of central users whose tweets were amplified by other users in the group and also across the network. The users at the centre of these groups downplayed the risk of obtaining a blood clot from a vaccine; these tweets were amplified within these broadcast groups. Group 3, on the other hand, is termed an ‘isolates’ group, in which there are no retweets or ‘@’ mentions. This group represents Twitter users who tweeted using the term ‘blood clot’ but did not mention or reply to any other user. We focused specifically on providing insight into the most popular tweets in Groups 1 and 2 because of their centralised discussion and overall impact on the network. Group 4 is a self-contained broadcast group; users in this group had little interaction with other users. We can describe this as an ‘echo chamber’. 

The most retweeted tweets in each of the two largest clusters were identified. In Group 1, the most retweeted message was as follows: 


*“So the odds of getting a blood clot from the J&J vaccine are literally one in a million and they stopped administering it. Do you know the odds of getting blood clots from birth control? It’s 1/1000. (This is considered very low odds)”.*


The most retweeted tweet in Group 2 was as follows: 


*“3 key points for folks recently vaccinated with J&J*



*1. your risk of getting hit by lightning this year > your risk of developing blood clot from vaccine*



*2. Don’t ignore a new severe headache, abd pain, or shortness of breath*



*3. Enjoy being vaccinated against a deadly disease.”*


Table 1 provides an overview of the key websites that were shared (i.e., included in tweets) on Twitter during this time. 

It can be seen that a range of websites reporting on general news was shared. These web sources were mainly based on news updates reporting on the links between blood clots and vaccines from countries around the world, such as the United States and Canada. However, not all the sources were scientific in nature, as the second most frequently shared link was an opinion piece by the *Washington Examiner*.

Table 2 provides an overview of the most frequently used hashtags.

The five most frequently used hashtags (aside from Covid-19) were #johnsonandjohnson (*n* = 260), #vaccine (*n* = 118), #astrazeneca (*n* = 85), #breaking (*n* = 53) and #covidvaccine (*n* = 33). Overall, the hashtags were used to have discussions about the blood clots reported in certain vaccines. It is not surprising to see ‘johnsonandjohnson’ and ‘astrazeneca’ appear among the most commonly used hashtags because news reports and discussions on Twitter frequently mentioned them.

Table 3 provides an overview of the most frequently used word pairs extracted from the tweets. Word pairs are a useful method to identify the topics of the conversations that took place. 

As shown in Table 3 above, the Twitter users conversed about blood clots in relation to the vaccine and co-words such as ‘blood, clot’, ‘clot, vaccine’, ‘blood, clots’ and ‘johnson, johnson’ appeared. These discussions were related to the general situation. A further group of keywords discussed the odds of suffering from a blood clot and utilised keywords such as ‘odds, blood’, ‘chance, blood’, ‘1 in 250,000, chance’, ‘control, 1 in 1000’ and ‘developing, blood’. The Twitter users also discussed the higher chance of suffering from a blood clot due to birth control, and this is reflected in Table 3 through the ‘birth- control’ word pair. 

Table 4 below provides an overview of the types of users that drove the discussion on Twitter during this time. 

Several influential users across disciplines, including writers, physicians, the general public, academics, celebrities and journalists, were influential in driving the key narratives and popular messages.

## 4. Discussion

Social media platforms are sources of health information for some members of the public and can have considerable influence on health decision making. Effective routes out of the current pandemic include large-scale vaccinations, and public confidence in vaccines plays an important role in their uptake. Stories concerning vaccines potentially causing blood clots became major news items and were discussed frequently on Twitter. To the best of our knowledge, this is the first study to analyse the discussions that have taken place on social media around COVID-19 vaccination and, specifically, the associated risk of thrombosis.

Our study sought to develop an understanding of the key narratives, groups and content within the network. It was found that the network had a high retweet ratio and resembled a broadcast network with high amplification. The two most popular tweets within Groups 1 and 2 served to encourage vaccinations and appeared to downplay the risk of blood clot events occurring. These discussions were driven by influencers across a broad range of disciplines. In our globalised world, social network debates may influence the behaviour of individuals and communities. 

An interesting finding was that the majority of the influential users and key websites were not from the world of scientific medicine. Instead, it was found that the majority of the information was provided by non-medical sources. An implication of this is that general scientific studies such as peer-reviewed papers, and other scientific health information are converted into content that is easily consumable by the general public, academics, celebrities and journalists. Our study highlights how these stakeholders played a positive role in sharing factual information. This is because the largest groups within the network shared content that downplayed the risk of suffering from a blood clot and encouraged other users to receive their vaccines. The response was community-driven and decentralised. 

Other studies [23,24] have noted the potential of social media for spreading disinformation and misinformation. However, our study highlighted a positive aspect of social media and its role in educating the general public with timely and factual information. Our network shape, outlined in Figure 1, highlighted how a few key and strategically placed users had the most impact on Twitter during this time. 

In our study, we found that social networks reinforced the global public health vaccination campaign. We recommend that healthcare stakeholders continue to be involved in the communication of scientific evidence on social media. Our results show an unintended positive message in favour of COVID-19 vaccination driven by non-healthcare influencers. The challenge to healthcare stakeholders is becoming part of this network. As one study suggests [25], for COVID-19, social media can have a crucial role in disseminating health information and tackling infodemics and misinformation. 

Health authorities and governments can take two lessons from this research: (1) Social network analysis and keyword searches of niche topics can help to identify sources of information, which can then be used for intelligence purposes, such as to counteract narratives that may contain dangerous disinformation. (2) Mainstream media and influential public figures can serve as valuable sources of information on social networks. These users have unique audiences and are able to exert considerable reach. These users can enhance public awareness during health crisis events. 

This study has several limitations. One limitation is the fact that Twitter is just one of the social networking services available, and other popular services, such as Facebook and Instagram, were not analysed. Another of the limitations of our study is that we were unable to relate the tweets and retweets to their geographical area. If this were possible, we could relate the social network activity in a certain geographical area with the coverage of vaccination in this area during a certain period. However, our results are likely to represent content that was commonly read and shared in English-speaking countries because our keywords to retrieve the data were in English and because the United States has the most Twitter users. Our focus on English-language tweets is a limitation of our study. We encourage researchers to proceed in this line of research to understand the effectiveness of tweets amplified across different languages and specific regions. However, our present results are likely to be of interest to stakeholders working in this area.

## 5. Conclusions

Twitter was used to highlight the low potential of developing a blood clot from vaccines and encouraged vaccinations among the public. Our results are likely to be of interest to health authorities, governments and stakeholders that are involved in vaccination programmes. 

## Figures and Tables

**Figure 1 ijerph-19-04584-f001:**
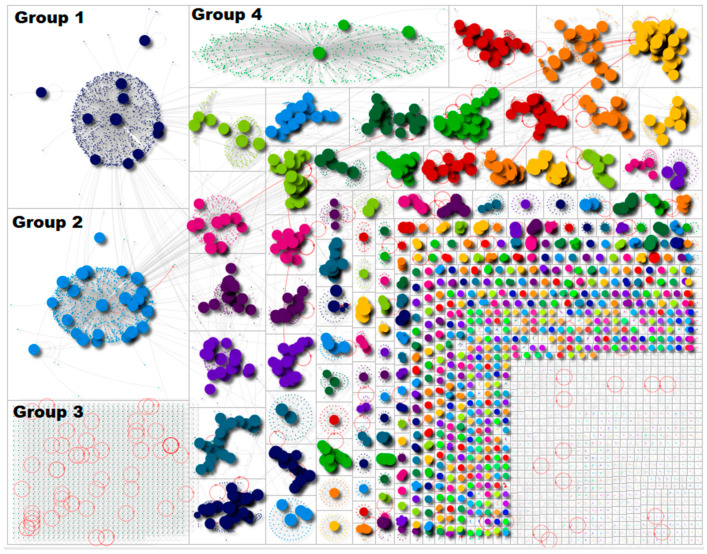
Social network analysis of the keyword ‘blood clots’ from 14 March to 14 April 2021.

**Table 1 ijerph-19-04584-t001:** Overview of key websites.

Title	Top URLs in Tweet in Entire Graph	No. of Times Shared
CDC, FDA To Review J&J Shot After 6 Blood Clot Cases Reported Out Of Nearly 7M Doses	https://www.npr.org/sections/coronavirus-live-updates/2021/04/13/986709618/u-s-recommends-pausing-use-of-johnson-johnson-vaccine-over-blood-clot-concerns?utm_source=twitter.com&utm_medium=social&utm_term=nprnews&utm_campaign=npr (accessed on 8 April 2022)	222
He won’t allow over-the-counter birth control, but Biden is pushing risky at-home abortions	https://www.washingtonexaminer.com/opinion/he-wont-allow-over-the-counter-birth-control-but-biden-is-pushing-risky-at-home-abortions (accessed on 8 April 2022)	93
If you’ve recently had the J&J vaccine, watch for these rare symptoms, CDC says	https://edition.cnn.com/2021/04/26/health/blood-clots-johnson-johnson-vaccine-wellness/index.html (accessed on 8 April 2022)	83
Link to USA FDA Tweet	https://twitter.com/US_FDA/status/1381925612743499778 (accessed on 8 April 2022)	82
Blood clot risks: comparing the AstraZeneca vaccine and the contraceptive pill	https://theconversation.com/blood-clot-risks-comparing-the-astrazeneca-vaccine-and-the-contraceptive-pill-158652 (accessed on 8 April 2022)	82
Clot questions over AstraZeneca and J&J vaccine	https://news.yahoo.com/clot-questions-over-astrazeneca-j-204253695.html (accessed on 8 April 2022)	75
US calls for pause in Johnson & Johnson vaccinations over blood clot concerns	https://abcnews.go.com/US/us-calls-halt-johnson-johnson-vaccination-blood-clot/story?id=77040882&cid=social_twitter_abcn (accessed on 8 April 2022)	65
CDC, FDA To Review J&J Shot After 6 Blood Clot Cases Reported Out Of Nearly 7M Doses	https://choice.npr.org/index.html?origin=https://www.npr.org/sections/coronavirus-live-updates/2021/04/13/986709618/u-s-recommends-pausing-use-of-johnson-johnson-vaccine-over-blood-clot-concerns (accessed on 8 April 2022)	57
CDC and FDA recommend US pause use of Johnson & Johnson’s COVID-19 vaccine over blood clot concerns	https://www.cnn.com/2021/04/13/health/johnson-vaccine-pause-cdc-fda/index.html (accessed on 8 April 2022)	49
Canadian public health agency confirms first report of blood clot linked to AstraZeneca	https://nationalpost.com/news/canadian-press-newsalert-phac-receives-report-of-blood-clot-linked-to-astrazeneca (accessed on 8 April 2022)	42

**Table 2 ijerph-19-04584-t002:** Overview of key hashtags.

Hashtag	Count
johnson and johnson	260
COVID-19	195
vaccine	118
astrazeneca	85
breaking	53
covidvaccine	33
coronavirus	33
covid	25
COVID-19 on tario	24
WSJ whats now	18

**Table 3 ijerph-19-04584-t003:** Overview of word pairs.

Top Word Pairs in Tweet in Entire Graph	h Count
blood, clot	13,709
birth, control	3581
clot, vaccine	3503
odds, blood	2911
blood, clots	2620
chance, blood	2030
johnson, johnson	1798
250,000, chance	1764
control, 1000	1491
Developing, blood	1435

**Table 4 ijerph-19-04584-t004:** Overview of Influential Users within the network.

Rank	Descriptionked by Betweenness Centrality
1	Writer
2	Physician
3	Member of Public
4	US Food and Drug Administration
5	Member of Public
6	National Public Radio (United States)
7	Academic
8	Cable News Network (CNN)
9	Celebrity
10	Journalist

## Data Availability

Not applicable.

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
