# Peer review of "A Social Network Analysis of Twitter Data Related to Blood Clots and Vaccines"

_ijerph, 2022, doi:10.3390/ijerph19084584_

Round 1

Reviewer 1 Report

I really appreciate this paper: The authors utilize Twitter to analyze how certain societal and health-related issues are perceived by the general public. As such the paper moves beyond self-reports and allows for an understanding of 'sensitive' issues (e.g. vaccination hesitancy during the covid-19 pandemic). The paper thus certainly has its merits, but I have a few comments for the authors to consider:

  • The aim of the study becomes clear to the reader quite late into the introduction, and even when it is introduced, it still remains quite vague what the authors intend to study -- merely how Twitter users discuss the issue of blood clots/ thrombosis and covid-19 vaccination? Or whether these discussions center around (or are framed) negative or positive perceptions surrounding vaccination, and how the news about possible blood clots factors into that perception? In the discussion (page 6 lines 162-163) this aim is more precisely formulated than anywhere in the introduction.
  • Relatedly, the introduction is written in a tone of voice that suggests that the authors are opinionated about the topic (namely that there is a risk for blood clots after vaccination with AstraZeneca or J&J). I think the authors should present their research aim(s) in the context of vaccination hesitancy more than they currently do, rather than to diverge on the back-and-forth between health agencies across the world during the discussions about (not) using these 2 vaccines. 
  • To me, it is entirely unclear how the data were collected/extracted and why only the words 'blood clots' were used, rather than e.g. 'thrombosis' or a combination of terms. Also, I assume that the authors focused on English-language tweets? The inclusion of Spanish tweets (for example) could provide the authors with more information about the issue they raise in the discussion section (page 6, lines 191-194). It is also unclear why they did not choose to focus on Spanish tweets but rather focused on English ones.
  • Terminology like 'Clauset-Newman-Moore algorithm' and 'betweenness algorithm' should be explained to the reader - what analysis do these provide?
  • The results are hard to understand - the figure is nice but it shows 4 major groups that are not discussed by the authors in the sense that group 3 is left out of consideration entirely, and that groups 1,2 and 4 were 'downplaying the risk' for getting blood clots after vaccinations. There were examples of most retweeted tweets for groups 1 and 2, but not for groups 3 and 4. -->All in all, I feel the authors need to take into account that most of the readers will not have any knowledge about social network analysis and will not understand their procedures and findings.
  • The authors seem overly optimistic in stating that their findings may be of use to health organizations and governments - in part because they do not provide sufficient details to determine the value of these findings, and partly because I don't think the shared information was as 'factual' as the authors state (page 6, line173-174). The 2nd source mentioned in Table 1 seems to be an opinion piece (anti-democrat?).

Author Response

Thank you for your comments, they have greatly helped us to improve the manuscript. See our responses in the attached document.

Reviewer 2 Report

This study is an interesting one analyzing the phenomenon of sharing and spreading of information about side effects (thrombosis) caused by vaccination through social media. The analysis was done through the SNA technique, and the results show that 4 groups were the dominant groups in sharing information related to vaccine side effects. It also presents the key arguments of the major groups.

However, despite the significance of this study, there are several limitations as follows, and supplementation or defense is required for these limitations.

First, it is necessary to emphasize the distinction between this study and previous studies. Various studies related to Covid-19 and social media have been conducted. Therefore, it is necessary to summarize the existing research and present the differentiation of this study through a separate chapter.

This study is limited to showing how information sharing and diffusion occurs through Social Network Analysis. In other words, it is just a description of how and in what group information is being shared.

If so, more detailed explanations of what kind of effect and influence such information sharing and dissemination have, the characteristics of the two leading groups shown through SNA, and how information is shared and disseminated will need to be explained in more detail.

It is representative of the data sample. The data used in this study were retrieved from Twitter. However, as for SNS, there are various SNS such as Facebook and Instargram as well as Twitter, and it is necessary to explain the representativeness of the data derived from Twitter.

The data sample of this study is retweets (n = 9,900) and tweets (n = 1, 952) are thought to have different information or interest in side effects (thrombus). Therefore, it is necessary to try the analysis by classifying it by type.

Overall, the analysis results are descriptive. The analysis results presented in each table (Tables 1 to 4) only describe the contents of the table. A detailed interpretation of the meaning of the results shown in the table needs to be added.

Author Response

(The authors gave the same response as above.)
